

# CapPlant: a capsule network based framework for plant disease classification

Omar Bin Samin[1], Maryam Omar[2] and Musadaq Mansoor[1,2]

[1] Center for Excellence in IT, Institute of Management Sciences (IMSciences), Peshawar, Peshawar, Pakistan

[2] Computer Science Department, National University of Computer and Emerging Sciences, Islamabad, Peshawar, Pakistan

## ABSTRACT

Accurate disease classification in plants is important for a profound understanding of their growth and health. Recognizing diseases in plants from images is one of the critical and challenging problem in agriculture. In this research, a deep learning architecture model (CapPlant) is proposed that utilizes plant images to predict whether it is healthy or contain some disease. The prediction process does not require handcrafted features; rather, the representations are automatically extracted from input data sequence by architecture. Several convolutional layers are applied to extract and classify features accordingly. The last convolutional layer in CapPlant is replaced by state-of-the-art capsule layer to incorporate orientational and relative spatial relationship between different entities of a plant in an image to predict diseases more precisely. The proposed architecture is tested on the PlantVillage dataset, which contains more than 50,000 images of infected and healthy plants. Significant improvements in terms of prediction accuracy has been observed using the CapPlant model when compared with other plant disease classification models. The experimental results on the developed model have achieved an overall test accuracy of 93.01%, with F1 score of 93.07%.

## INTRODUCTION

The existence, survival and development of human race revolve around agriculture, as the major portion of food is derived from agriculture. The modern technological agriculture sector strives to enhance the quality and production of farming products and coping with cultivation diseases. These diseases are a major threat to agricultural development as they adversely affect plants growth and quality, resulting in reduced crop yield (*Yin & Qiu, 2019*). To minimize such threats, the complex and unpredictable agricultural ecosystem requires continuous monitoring to analyze diverse physical and environmental aspects. Deep Leaning (DL) can be utilized as it constitutes a modern and state-of-the-art technique for image processing and data analysis with great potential and promising results (*Kamilaris & Prenafeta-Boldú, 2018*). DL has been successfully applied in various domains like healthcare (*Miotto et al., 2018*), automatic machine translation (*Singh et al., 2017*), automatic text generation (*Pawade et al., 2018*), image recognition (*Satapathy et al., 2019*) and agriculture (*Giménez-Gallego et al., 2020*; *Zheng et al., 2019b*), etc.

Corresponding author
Omar Bin Samin,
omar.samin@imsciences.edu.pk

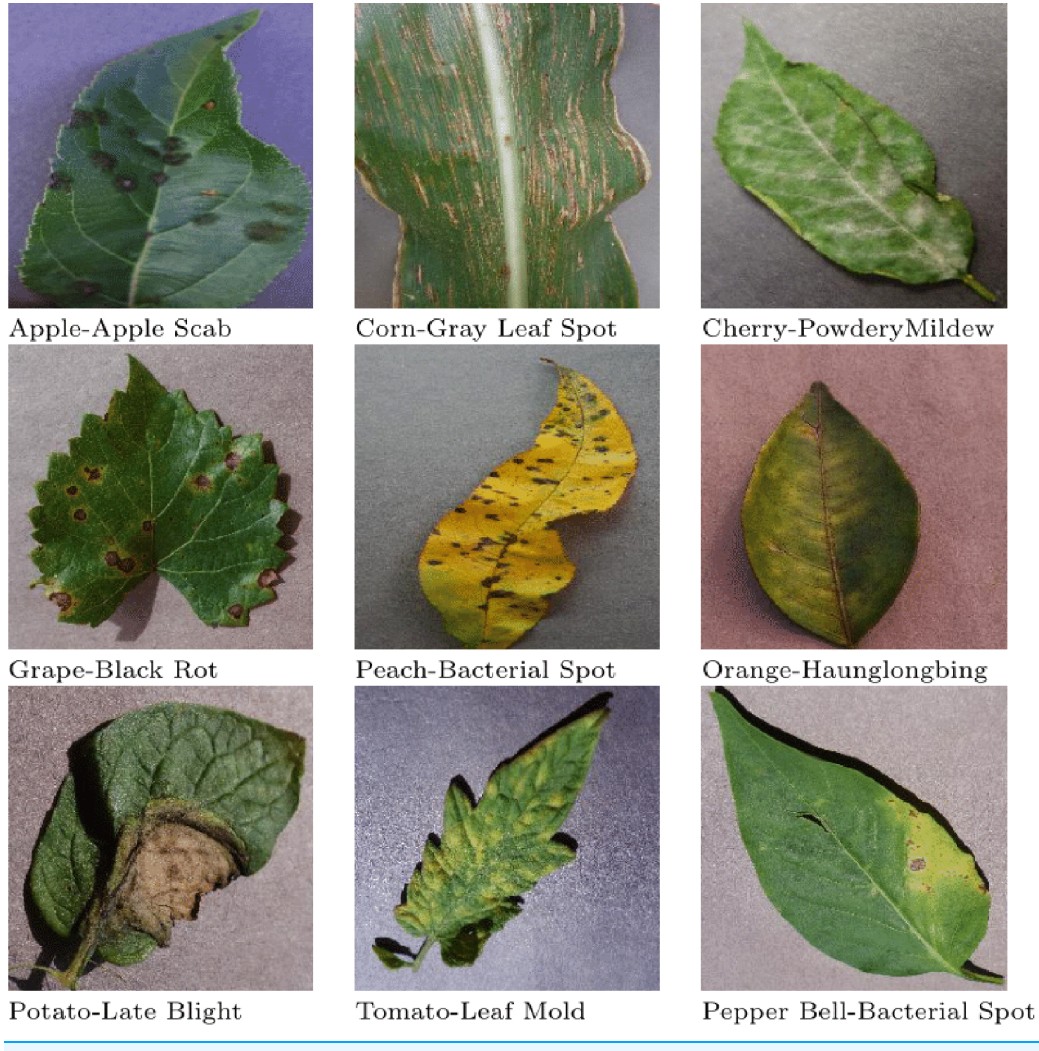

Apple-Apple Scab     Corn-Gray Leaf Spot     Cherry-PowderyMildew

Grape-Black Rot     Peach-Bacterial Spot     Orange-Haunglongbing

Potato-Late Blight     Tomato-Leaf Mold     Pepper Bell-Bacterial Spot

**Figure 1 Images of diseases found in various plants.**

In the past few years, researchers have targeted many crops such as blueberry, wheat, tomato and cherry for classification. Moreover, they have also targeted many plant diseases like leaf mold, late bright, tomato mosaic virus disease, two-spotted spider mite attack, target spot, tomato yellow leaf curl virus, rust, tan spot, septoria and others for detection. Figure 1 shows images of different diseases found in various plants.

Previously, researchers utilized handcrafted features along with classifiers for solving plant disease classification problems (*Salazar-Reque et al., 2019*; *Shruthi, Nagaveni & Raghavendra, 2019*). Presently, due to success of DL techniques, many researchers are using them for solving various classification problems (*Picon et al., 2019a*; *Ferentinos, 2018*; *Picon et al., 2019b*; *Too et al., 2019*; *Kamal et al., 2019*). Pre-trained models such as AlexNet, GoogleNet, DenseNet, MobileNet and VGG16net, etc. are widely used for plant disease classification. Although they have shown some promising results, however their time complexity and computational cost still needs improvement (*Picon et al., 2019b*; *Too et al., 2019*; *Kamal et al., 2019*). Researchers have also created their own custom

Convolution Neural Network (CNN) based models for classification and tested on various datasets (*Picon et al., 2019a*; *Ferentinos, 2018*). These models have some major drawbacks, for instance, one major issue with some of these models is in targeting less number of crops and diseases, secondly they are presenting results using limited or non-standard evaluation metrics like Accuracy (Testing, Training and Validation), Precision, Recall and F1-Score that are generally used for evaluating a classification model. In this research, a deep learning architecture (CapPlant) is developed using CNN along with capsule network to classify and detect any disease found in plants accurately.

Capsules in a capsule network (*Sabour, Frosst & Hinton, 2017*; *Hinton, Krizhevsky & Wang, 2011*; *Hinton, Sabour & Frosst, 2018*) are the groups of neurons that encode spatial information as well as the probability of an object being present. In contrast to CNN, capsule encodes information in a vector form to store spatial information as well. In recent years, capsule networks have been used for detection (*Afshar, Mohammadi & Plataniotis, 2018*), text classification (*Zhao et al., 2018*; *Kim et al., 2020*; *Zhao et al., 2019*; *Ren & Lu, 2018*), tumor classification (*Afshar, Plataniotis & Mohammadi, 2019a*; *Afshar, Plataniotis & Mohammadi, 2019b*), bioinformatics (*de Jesus et al., 2018*) and simple classification problems (*Lukic et al., 2019*; *Hilton et al., 2019*; *Zhao et al., 2018*). Keeping in view success of capsule network, (*Bass et al., 2019*; *Jaiswal et al., 2018*; *Upadhyay & Schrater, 2018*) have also explored capsule with Generative Adversarial Networks(GANs) and have presented some promising results.

Conceptual novelty of this work is the utilization of capsule network along with CNN. As CNN stores information in scalar form, they are considered as translational and rotational invariant, whereas in the capsule network, information is grouped together in form of vector where the length of a capsule vector represents the probability of the existence of a feature in an image and the direction of the vector would represent its pose information.Therefore, exploiting capsule layer enables the model to capture relative spatial and orientation relationship between different entities of an object in an image.

The rest of the paper is organized as follows: "Related Work" briefly explores different studies related to plant disease detection; implementation details along with models on which CapPlant is built upon are discussed in detail in "Methodology" followed by results in "Result". The paper is concluded in "Conclusion", along with some future recommendations.

## RELATED WORK

A considerable amount of literature has been published on plant disease classification using conventional machine learning (*Salazar-Reque et al., 2019*; *Shruthi, Nagaveni & Raghavendra, 2019*) and deep learning (*Picon et al., 2019b*, *Too et al., 2019*; *Kamal et al., 2019*) techniques. Few of the existing research have also explored capsule network for plant disease classification (*Kurup et al., 2019*; *Li et al., 2019*). This section explains a few notable previous research work on plant disease classification.

*Sullca et al. (2019)* implemented diseases detection in blueberry leaves using computer vision and machine learning techniques. Noise removal in images was handled with the help of gaussian blur and median blur filters. Details in each image were enhanced with the help of weighted filters. Blueberry leave pictures were then group into three categories:
plagued, healthy and diseased. Local binary patterns and histogram of oriented gradients were used for characteristics extraction. Due to unavailability of blueberry leaves dataset, a custom dataset was created. Deep learning model gave an accuracy of 84%, predicting the disease of blueberry leaves. The proposed system only considered blueberry pest infections.

*Ashqar & Abu-Naser (2018)* implemented tomato leaves diseases detection using DL. Among many diseases that can exist in tomato plant, only five were considered. CNN used for classification was divided into two parts. The first part used for feature extraction consisted of four convolution layers along with activation function ReLU followed by max pooling, while the second part of the model comprised of two dense layers followed by flattening layer. Softmax was used as an activation for the second part. Experiments were conducted on two type of images; one with three color channels and other with one color channel. Nine thousand healthy and infected tomato images were considered for training.The dataset was created for six classes which included early blight, bacterial spot, septorial leaf spot, leaf mold, bacterial spot, yellow leaf curl virus and healthy. Original images were resized to a smaller size of 150 * 150 so that computation could be faster. Quality of images were maintained so that disease detection could work well. The proposed model gave an accuracy of 99.84% on three color channels, whereas achieved accuracy was 95.54% on one color channel. The data collection process was manual and tedious, which resulted into considering limited number of diseases.

*Salazar-Reque et al. (2019)* implemented an algorithm for detecting visual symptom in plants disease. The images used were grouped into nine different categories by diseases and plants.These nine groups consisted of seven plants *i.e.*, apple, grape, mango, potato, quinoa, peach and avocado. The target diseases for apple were scab, cedar apple rust and black rot. Target disease for avocado was necrosis and infection, target disease for grape was black rot, target disease for mango was necrosis, target disease of potato was alternaria, target disease for peach was bacterial spot and target disease for quinoa was mildew. Total 279 images were considered, out of which 90 belonged to apple diseases, 30 belonged to grape disease, 30 belonged to avocado, 30 belonged to mango disease, 30 belonged to potato disease, 30 belonged to peach disease and 30 belonged to quinoa. No image of healthy plants were considered. The system developed used a clustering algorithm for putting together same color pixels in regions known as super pixels. A total of 279 pictures of leaves were used. The proposed system used no images of healthy plants and a small dataset of infected plants images. The images groups were giving different True Positive Rate (TPR) and False Positive Rate (FPR), indicating proper groups were not formed.

*Reddy et al. (2019)* presented an idea of combining bioinformatics with image processing for detecting diseases in crops and plants. In the proposed methodology, HSI (Hue, Saturation, Intensity) algorithm for image segmentation was used. A digital camera was used for capturing image and unwanted areas of image were removed using different techniques. The pixels in which the value of green intensity were more than the desired threshold were, considered as unhealthy crops/plants. The authors did not mention any detail about the dataset they used for the experiment. The authors also failed to mention any results based on which they achieved the stated conclusions.

*Picon et al. (2019b)* implemented CNN model for classification of plant diseases. Three CNN models were proposed for combining different aspects together like crop identification data, geographical locations and weather conditions etc. Around one hundred thousand pictures of the actual field conditions were taken from the cell phone. Dataset was created for seventeen diseases of five crops. The crops included winter wheat, corn, rapeseed, winter barley and common rice. The diseases considered were *Septoria tritici, Puccinia striiformis, Puccinia recondite, Septoria nodorum, Drechslera triticirepentis, Oculimacula yallundae, Gibberella zeae, Blumeria graminis, Helminthosporium turcicum, Phoma lingam, Pyrenophora teres, Ramularia collo-cygni, Rhynchosporium secalis, Puccinia hordei*, various diseases, *Thanatephorus cucumeris, Pyricularia oryzae*. Total 1,20,950 images were considered, out of which 11,295 belonged to common rice, 32,229 belonged to winter barley, 13,774 belonged to rapeseed, 64,026 to winter wheat and 631 belonged to corn. Independent particular crop models reported a balance accuracy of 0.92, whereas multi crop reported a balance accuracy of 0.93. The proposed system lacked in sharing complete training and testing results. The system lacked in considering number of result variants *i.e.*, F1 score, recall and precision.

*Picon et al. (2019a)* implemented deep convolution networks for disease detection in crop. The images used were divided into four groups of Rust, Tan Spot, Septoria and Healthy. The images were taken from Wheat 2014; Wheat 2015 and Wheat 2016 databases. Total 8,178 images were considered, out of which 3,338 belonged to Rust, two thousand seven hundred and 44 belonged to Septoria, 1,568 belonged to Tan spot, 1,116 belonged to Healthy class. A total of 1,385 images were taken from Wheat 2014 database, 2,189 images were taken from Wheat 2015 and 3,969 images were taken from Wheat 2016 database. The proposed technique used residual neural networks with many improvements in tile cropping and augmentation scheme. A mobile application was developed for providing input to the system; pictures were taken manually from the app and then loaded on a server for further processing. More than 8,000 images were considered for training. The system was able to process the image and find the disease in quick time. The implemented technique considered only three diseases for wheat crop, which made the scope minor.

*Kamal et al. (2019)* implemented depth wise separable architectures (convolution) for classification of plant diseases. The system developed used leaves images for detection of plant diseases. Several models were trained using the proposed method and reduced MobileNet stood out. More than 80,000 images dataset was considered for training and testing, which covered 55 classes of healthy and diseased plants. The images were taken from PlantVillage for training and for testing PlantLeafs dataset was used. A total of 82,161 images of PlantVillage were considered, 18,517 images of PlantLeaf1 were considered, 23,110 images of PlantLeaf2 were considered and 32,241 images of PlantLeaf3 were considered. The number of classes included in PlantVillage are 55, whereas in PlantLeaf1 these are 18, in PlantLeaf2 these are 11 and in PlantLeaf3 these are 16. It gave 36.03% accuracy when tested on pictures taken under different parameters than those of training. Even though the number of image dataset of healthy/diseased plants were more, nonetheless the developed system only considered accuracy as an evaluation metric and reported no precision, recall and F1 score.

*Sengar, Dutta & Travieso (2018)* implemented identification and quantification of powdery mildew disease in cherry using computer vision based technique. Adaptive intensity focused thresholding method was proposed for powdery mildew disease automatic segmentation. Two parameters were used in assessment of the level of disease spread in plants: (1) the portion in plant that was effected by the disease and (2) the length of the effected portion in plant. The proposed model achieved 99% accuracy. The proposed technique may be used for predicting only one disease in cherry plant.

*Rangarajan, Purushothaman & Ramesh (2018)* implemented tomato disease classification with the help of pre-trained deep learning algorithm. Two pre-trained models *i.e.*, VGG16net and AlexNet were used by the authors. A total of 13,262 tomato images from PlantVillage (40) dataset containing six disease classes and one heathy class were used by the proposed system. Accuracy reported for disease classification using VGG16net was 97.29% and using AlexNet was 97.49%. Comparing AlexNext and VGG16net, minimum execution time and better accuracy were reported with AlexNet. The authors considered six diseases for tomato plant, for which they used pre-trained networks.

*Mohanty, Hughes & Salathé (2016)* implemented plant disease detection using DL techniques on plant images All images used were resized $256 \times 256$, both model prediction and optimizations were performed on these images. A total of 26 diseases in fourteen crops were detected using this model. Pre-trained models, AlexNet and GoogleNet were considered for this experiment. Models were trained on three different variations of PlantVillage datasets, first they were trained on color images, then on gray scale images and finally on segmented leaves images. A dataset containing 54,306 images was used, containing healthy and diseased plant leaves. The dataset targeted different 38 classes. Five different training–test distributions were used *i.e.*, first train 80%–test 20%, second train 60%–test 40%, third train 50%–test 50%, fourth train 40%–test 60%, last one train 20%–test 80%. Color, grey scale and leaf-segmented images were considered. Two different training mechanisms were considered, first transfer learning and second training from scratch. The model achieved 99.35% accuracy, but on a held out test set. The system dropped accuracy to 31% when tested on different images other than training images. The developed technique used pre-trained networks instead of developing their own neural network for classification.

*Barbedo (2019)* implemented plant disease identification using deep learning. The diagnosis in the given algorithm considers image classification on two things, one spots and second lesions. A total of 46,409 images were considered for disease identification. The images were taking using many sensors. The resolution of captured images were upto 24 MPixels. The plants considered were common bean, cassava, citrus, cocunut tree, corn, Kale, Cashew Tree, Coffee, Cotton, Grapevines, Passion fruit, Soybean, Sugarcane and wheat. Overall, 14 plants and 79 diseases were considered, but many had very few images associated with them. The model used was pretrained GoogLNet CNN. Accuracy was reported for different plants. The developed technique used pre-trained networks instead of developing their own neural network for classification. The developed system focused more on creating a custom dataset for disease detection. The developed system used fewer images for many classes. Many conditions had a few images associated with them in the dataset captured.

*Durmus, Günes & Krc (2017)* implemented tomato disease detection using deep learning. Diseases that occurred in tomato fields or greenhouses both were considered. AlexNet and SqueezeNet algorithms were used for training and testing of tomato disease detectio. Images were taken from PlantVillage dataset. Only tomato images were considered. The diseases considered were leaf mold, spider mites, septoria leaf spot, early blight, bacterial spot, mosaic virus, yellow curl virus and target spot. Accuracy reported for tomato disease classification using AlexNet was 95.65% and using SqueezeNet was 94.3%. The authors considered only tomato diseases and instead of creating their own neural network use pre-built models. The authors failed to report other evaluation metrics like F1 score, precision or recall.

*Ramcharan et al. (2017)* implemented cassava disease detection using deep learning. The dataset was custom build using Sony Cybershot 20.2 mp camera. The dataset was captured over a period of 4 weeks and consisted of about 11,670 images. The dataset was named "leafleft cassava dataset". Five diseases were considered that are Cassava brown streak disease, Red mite damage, Cassava mosaic disease, Green mite damage and Brown leaf spot. A deep convolutional neural network Inception v3 was used for cassava disease detection. The last layer of CNN was replaced with three different variations to test the model on three different architectures. Three different architectures were support vector machines, softmax layer and knn. Confusion matrix was reported for different cassava diseases. The proposed technique is used only for cassava plant. The proposed technique considered only five out of many diseases.

*Zheng et al. (2019a)* and *Kurup et al. (2019)* has explored capsule network and CNN for plant disease classification. Network architecture in (*Zheng et al., 2019a*) utilized two convolutional layers and one primary capsule layer for training and testing. However, the proposed model only presented test and train precision of 88% and 90% respectively. For reducing the drawbacks and also to get better performance, new architecture of CNN; capsulenet is implemented in (*Kurup et al., 2019*). Capsulenet was analyzed for two datasets: (1) first model was built for diagnosis of plant disease using plant leaves images. The dataset used for training contained 54,306 images of 14 different plant species. The proposed architecture reported an accuracy of around 94%. (2) second model was trained for classification of plant leaves. The dataset used for training had 2,997 images of 11 plants. The prediction model with capsulenet gave an accuracy of around 85%.

## Limitations of existing systems

All the above crop disease detection techniques work well for detecting the diseases in crops, however they have few limitations such as:

1. **Limited Scope:** Less number of crops/diseases are targeted.
2. **Limited Evaluation Metrics:** Conclusion has been achieved based on few result parameters.
3. **Limitation of CNN:** Most of the techniques used pre-trained networks or created their own CNN. The problem with CNN is that it only considers presence of entities in an object, it does not take into account not relative spatial relationship between them.

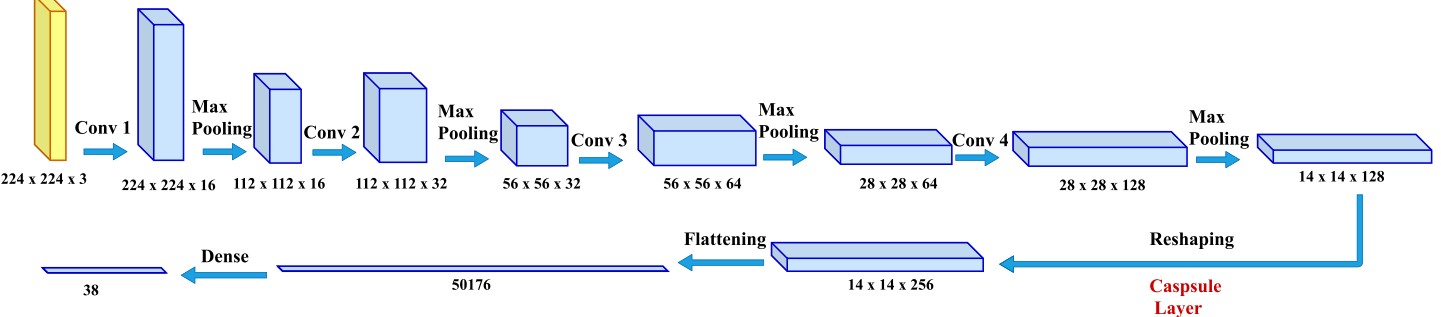

**Figure 2 Network architecture for CapPlant: deep learning architecture for plant disease prediction through pictures.** CapPlant is a real deep learning architecture because it uses end-to-end learning.

## METHODOLOGY

To predict plant diseases from the given images, simple yet effective model CapPlant is proposed in which the last convolutional layer is replaced by state-of-the-art capsule layer to incorporate relative spatial and orientational relationship between different entities of a plant in an image. The overall pipeline of the proposed model is illustrated in Fig. 2. The following subsection explains capsule network and the deep learning architecture of CapPlant that performs end-to-end learning for plant disease classification. End-to-end learning replaces a pipeline of components with a single deep learning neural network and goes directly from the input to the desired output. It eradicates the need of preprocessing or complex feature extraction process by learning directly from the labeled data and simplifies the decision making process. End-to-end models are of key importance for building artificial intelligence systems because of their simplicity, performance and data-driven nature (*Rafique, Fu & Mai, 2021*).

### Capsule network

In the field of DL, although CNN has been a huge success, however, they have some major drawbacks in their basic architecture which causes them to fail in performing some major tasks. CNN automatically extracts features from images, and from these features it learns to detect and recognize different objects. Early layers extract simple features like edges, and as layers proceed features become more and more complex. At the end, CNN uses all extracted features to make a final prediction. Here lies the major drawback in basic architecture of CNN that only presence of feature is captured and nowhere in this approach spatial information is stored.

Capsule Network (*Sabour, Frosst & Hinton, 2017*; *Hinton, Krizhevsky & Wang, 2011*; *Hinton, Sabour & Frosst, 2018*) have recently been proposed to address this limitation of CNN. Capsules are the groups of neurons that encode spatial information as well as the probability of an object being present. In capsule network, corresponding to each entity in an image, there is a capsule which gives:

1. Probability that the entity exists.
2. Instantiation parameters of that entity.

The main operations within capsules are performed as follows:

To encode the imperative spatial association between low and high level features within the image, the multiplication of the matrix of the input vectors with the weight matrix is calculated.

$$\hat{u}_{j|i} = W_{ij}u_i + B_j \tag{1}$$

The sum of the weighted input vectors is calculated to determine that the current capsule will forward its output to which higher level capsule.

$$su_j = \sum_i c_{ij}\hat{u}_{j|i} \tag{2}$$

Finally, non-linearity is applied using the squash function. While maintaining a direction of a vector, the squashing function maps it to maximum length of one and minimum length of 0.

$$v_j = squash(su_j) \tag{3}$$

## Network architecture

### Model inputs

Inputs to the network are plant images of size $224 \times 224 \times 3$. The size of inputs to CapPlant network is represented as (y; $224 \times 224 \times 3$), where y is the batch size. At the expense of reduced accuracy, small batch sizes lead to faster training. Relatively large batch sizes are used to increase accuracy at the expense of slower training. For training of CapPlant, batch size is set to 32.

### Feature representation layers

When the input is fed into CapPlant, features or representations is extracted from the inputs by passing it through four layers of convolution, followed by ReLU activation and max polling after each layer. The output of the 4th convolutional layer is reshaped and passed through the capsule layer to capture relative spatial and orientational relationship between different entities of an object in the image. The tensor obtained from the capsule layer is flattened and then passed through a densely-connected Neural Network (NN) layer. In the end, softmax is applied to squash a vector in the range (0, 1) such that all the resulting elements add up to one. For a particular class $c_i$, the softmax function can be calculated as follows:

$$f(c)_i = \frac{e^{c_i}}{\sum_j^C e^{c_j}} \tag{4}$$

where $c_j$ are the scores inferred by the model for each class in $C$. Softmax activation for a class $c_i$ depends on all the scores in $c$.

| input_1: InputLayer | Input: | (32, 224, 224, 3) |
|---|---|---|
| | Output: | (32, 224, 224, 3) |

| conv2d_1: Conv2D | Input: | (32, 224, 224, 3) |
|---|---|---|
| | Output: | (32, 224, 224, 16) |

| max_pooling2d_1: MaxPooling2D | Input: | (32, 224, 224, 16) |
|---|---|---|
| | Output: | (32, 112, 112, 16) |

| conv2d_2: Conv2D | Input: | (32, 112, 112, 16) |
|---|---|---|
| | Output: | (32, 112, 112, 32) |

| max_pooling2d_2: MaxPooling2D | Input: | (32, 112, 112, 32) |
|---|---|---|
| | Output: | (32, 56, 56, 32) |

| conv2d_3: Conv2D | Input: | (32, 56, 56, 32) |
|---|---|---|
| | Output: | (32, 56, 56, 64) |

| max_pooling2d_3: MaxPooling2D | Input: | (32, 56, 56, 64) |
|---|---|---|
| | Output: | (32, 28, 28, 64) |

| conv2d_4: Conv2D | Input: | (32, 28, 28, 64) |
|---|---|---|
| | Output: | (32, 28, 28, 128) |

| max_pooling2d_4: MaxPooling2D | Input: | (32, 28, 28, 128) |
|---|---|---|
| | Output: | (32, 14, 14, 128) |

| reshape_1: Reshape | Input: | (32, 14, 14, 128) |
|---|---|---|
| | Output: | (32, 14, 14, 1, 128) |

| conv_cap: ConvCapsule Layer | Input: | (32, 14, 14, 1, 128) |
|---|---|---|
| | Output: | (32, 14, 14, 1, 256) |

| reshape_2: Reshape | Input: | (32, 14, 14, 1, 256) |
|---|---|---|
| | Output: | (32, 14, 14, 256) |

| flatten_1: Flatten | Input: | (32, 14, 14, 256) |
|---|---|---|
| | Output: | (32, 50176) |

| dense_1: Dense | Input: | (32, 50176) |
|---|---|---|
| | Output: | (32, 38) |

**Figure 3 Visual illustration of data flow between each layer of the proposed CapPlant model.**

### Normalization and output layers

The CapPlant model is compiled based on the features extracted from the previous layers. To calculate the error, the Categorical Cross Entropy loss (CE) is used as follows:

$$CE = -log\left(\frac{e^{c_p}}{\sum_{j}^{C} e^{c_j}}\right) \tag{5}$$

where $c_p$ is the CNN score for the positive class.

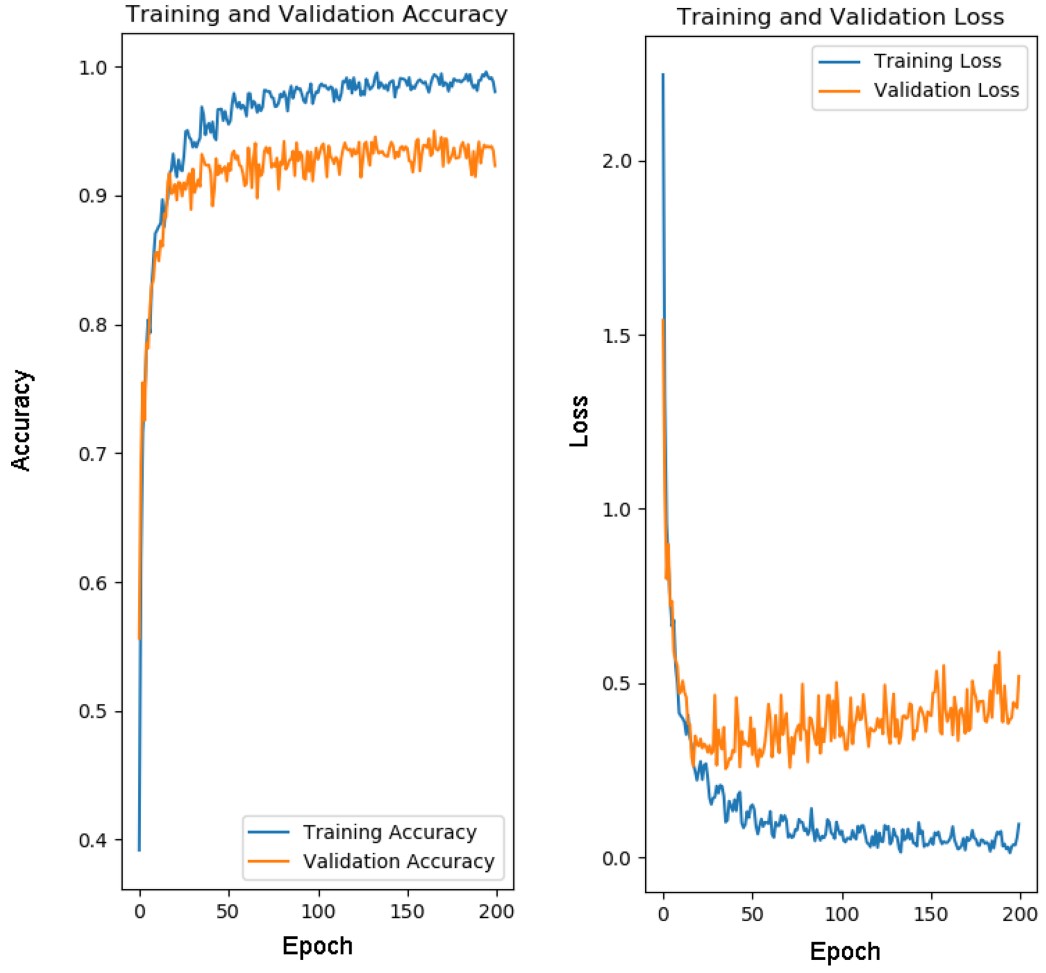

**Figure 4** **Learning curves for CapPlant.** To prevent overfitting of data, early stopping was employed at epoch 100.

## RESULT

This section presents the results of the CapPlant model. Extensive experimentation is carried out to evaluate the performance of our model.

### Experiments

#### *Experimental setup*

GPU Tesla K40c workstation is used as baseline system for training, testing and validation of the CapPlant model. Keras, OpenCV, Capsule Layers, Matplotlib and CuDNN libraries are used for software implementation of CapPlant.

#### *Training*

For training of our model CapPlant, Adam is applied as an optimizer, categorical cross entropy is utilized for calculating loss and accuracy is used as evaluation metric. Overall training and validation losses and accuracies are exploited to determine the performance of

**Table 1  Hyper parameters set for training of CapGAN.**

| Parameters | Value |
|---|---|
| Batch size | 32 |
| Epochs | 100 |
| Image size | 224 × 224 × 3 |
| Learning rate | 0.0002 |
| Momentum for Adam update | 0.5 |
| Loss | Categorical cross entropy |

**Table 2  Detail of each CapPlant layer along with output shape and number of obtained parameters.**

Total training images: 5,000
Total validation images: 5,423
Total validation images: 5,470
Total classes: 38
43429 Images belonging to 38 classes
5417 Images belonging to 38 classes
5459 Images belonging to 38 classes

| Layer | Output shape | Param # |
|---|---|---|
| Input layer | (32, 224, 224, 3) | 0 |
| Conv2D | (32, 224, 224, 16) | 448 |
| MaxPooling2 | (32, 112, 112, 16) | 0 |
| Conv2D | (32, 112, 112, 32) | 4,640 |
| MaxPooling2 | (32, 56, 56, 32) | 0 |
| Conv2D | (32, 56, 56, 64) | 18,496 |
| MaxPooling2 | (32, 28, 28, 64) | 0 |
| Conv2D | (32, 28, 28, 128) | 73,856 |
| MaxPooling2 | (32, 14, 14, 128) | 0 |
| Reshape | (32, 14, 14, 1, 128) | 0 |
| ConvCapsuleLayer | (32, 14, '4, 1, 256) | 295,168 |
| Reshape | (32, 14, 14, 256) | 0 |
| Flatten | (32, 50176) | 0 |
| Dense | (32,38) | 1,906,726 |

Total params: 2, 299, 334
Trainable Params: 2, 299, 334
Non-trainable params: 0

our model. Table 1 lists parameters along with their values adjusted for training and testing of CapPlant, whereas the, detail of each layer designed for testing, training and validation of CapPlant model is listed in Table 2. Furthermore, Fig. 3 is a visual illustration of data flow between each layer of proposed CapPlant model. Moreover, Fig. 4 shows the learning curves obtained while calculating training and validation accuracy and losses. The CapPlant model is trained for total 200 epochs, however, trained models at 50, 100, and 150 epochs were also obtained for the sake of comparison. Also, early stopping was employed at epoch 100 to avoid over fitting.

**Table 3 Details of PlantVillage dataset used for testing and training of CapPlant.**

| Plant | Class label | Name | # of Training samples | # of Validation samples | # of Testing samples |
|---|---|---|---|---|---|
| Apple | 0 | Apple scab | 504 | 63 | 63 |
| | 1 | Black rot | 496 | 62 | 63 |
| | 2 | Cedar Apple rust | 220 | 27 | 28 |
| | 3 | Healthy | 1,316 | 164 | 165 |
| Blueberry | 4 | Healthy | 1,201 | 150 | 151 |
| Cherry | 5 | Healthy | 683 | 85 | 86 |
| | 6 | powdery mildew | 841 | 105 | 106 |
| Corn | 7 | Gray leaf spot | 410 | 51 | 52 |
| | 8 | Common rust | 953 | 119 | 120 |
| | 9 | Healthy | 929 | 116 | 117 |
| | 10 | Northern leaf blight | 788 | 98 | 99 |
| Grape | 11 | black rot | 944 | 118 | 118 |
| | 12 | Esca black measles | 1,106 | 138 | 139 |
| | 13 | Healthy | 338 | 42 | 43 |
| | 14 | Leaf blight | 860 | 107 | 109 |
| Orange | 15 | Haunglonbing | 4,405 | 550 | 552 |
| Peach | 16 | Bacterial spot | 1,837 | 229 | 231 |
| | 17 | Healthy | 288 | 36 | 36 |
| Pepper Bell | 18 | Bacterial spot | 797 | 99 | 101 |
| | 19 | Healthy | 1,182 | 147 | 149 |
| Potato | 20 | Early blight | 800 | 100 | 100 |
| | 21 | Healthy | 121 | 15 | 16 |
| | 22 | Late blight | 800 | 100 | 100 |
| Raspberry | 23 | Healthy | 296 | 37 | 38 |
| Soybean | 24 | Healthy | 4,072 | 509 | 509 |
| Squash | 25 | Powdery Mildew | 1,468 | 183 | 184 |
| Starberry | 26 | Healthy | 364 | 45 | 47 |
| | 27 | Leaf scorch | 887 | 110 | 112 |
| Tomato | 28 | Bacterial spot | 1,701 | 212 | 214 |
| | 29 | Early blight | 800 | 100 | 100 |
| | 30 | Healthy | 1,272 | 159 | 160 |
| | 31 | Late blight | 1,527 | 190 | 192 |
| | 32 | Leaf Mold | 761 | 95 | 96 |
| | 33 | Septoria leaf spot | 1,416 | 177 | 178 |
| | 34 | Spider Mites | 1,340 | 167 | 169 |
| | 35 | Target spot | 1,123 | 140 | 141 |
| | 36 | Mosaic virus | 298 | 37 | 38 |
| | 37 | Yellow leaf curl virus | 4,285 | 535 | 537 |

## Results of experiments

**Dataset.** In this research, PlantVillage; An open access repository of images on plant health to enable the development of mobile disease diagnostics *Hughes & Salath'e (2015)* obtained

**Table 4 Comparison of different evaluation metrics measured for CapPlant with various state-of-the-art models.**

| Year | Model | Training accuracy | Validation accuracy | Test accuracy | Precision | Recall | F1-Score | Average |
|------|-------|-------------------|---------------------|---------------|-----------|--------|----------|---------|
| 2018 | VGG net | 83.86% | 81.92% | 81.83% | – | – | – | 82.53% |
| 2019 | Capsule network | – | – | – | 88% | – | – | 88% |
| 2020 | CapPlant | 98.06% | 92.31% | 93.07% | 93.07% | 93.07% | 93.07% | 93.77% |

from source (*Mohanty, 2018*) has been used for training and testing. PlantVillage dataset have 54,306 images belonging to 14 different plants. There are total 38 classes, out of which 26 depicts diseases from various plants, whereas 12 are classes of different plant with healthy leaves. The dataset is randomly split into train, validation and test sets containing roughly 80%, 10% and 10% of images respectively to avoid overfitting. The details of complete dataset is given in Table 3.

### Quantitative evaluation

For calculating predicting performance of CapPlant model, several evaluation metrics are calculated such as F1 score, accuracy, recall and precision.

$$accuracy = \frac{T_p + T_n}{T_p + T_n + F_p + F_n}$$

$$recall = \frac{T_p}{T_p + F_n}$$

$$precision = \frac{T_p}{T_p + F_p}$$

$$F_1 = 2 * \frac{precission * recall}{precission + recall}$$

where $T_p$ stands for true positive, $T_n$ for true negative, $F_p$ for false positive and $F_n$ for false negative. A $T_p$ is a result where the model predicts the positive class correctly. Likwise, a $T_n$ is a result where the model predicts the negative class correctly. A $F_p$ is a result where the model predicts the positive class wrong, and $F_n$ is an outcome where the model predicts the negative class wrong. Table 4 shows the values of above evaluation metrics calculated for CapPlant model. Figures 5 and 6 demonstrates bar chart representing recall, precission and F1 score, calculated for each disease and healthy category of plants respectively.

### Comparison with previous Models

To further demonstrate the effectiveness of the proposed CapPlant model, it is compared with previous state-of-the-art models for plant disease classification and detection. Table 4 lists various evaluation metrics calculated using different models for PlantVillage dataset and evidently shows that CapPlant outperforms previous models.

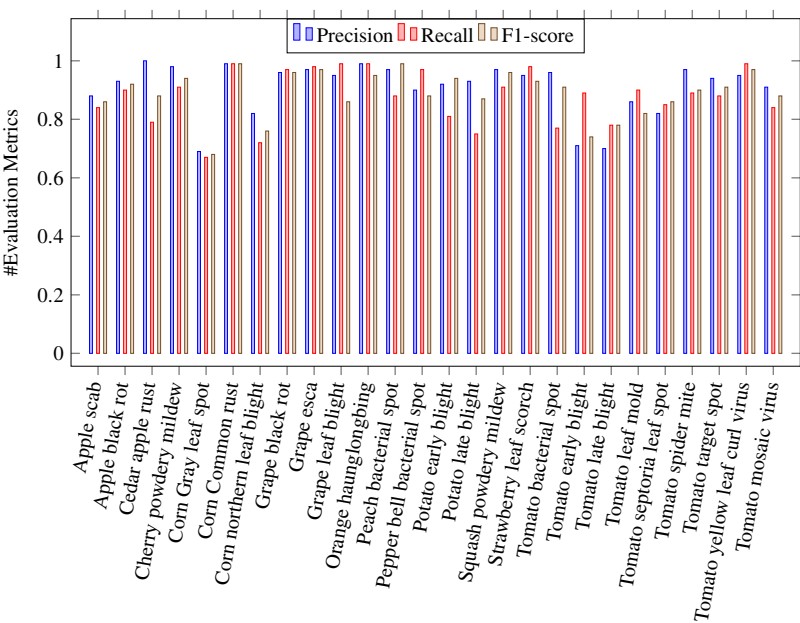

**Figure 5** Bar chart representing precision, recall and F1 score, calculated for each 26 plant diseases.

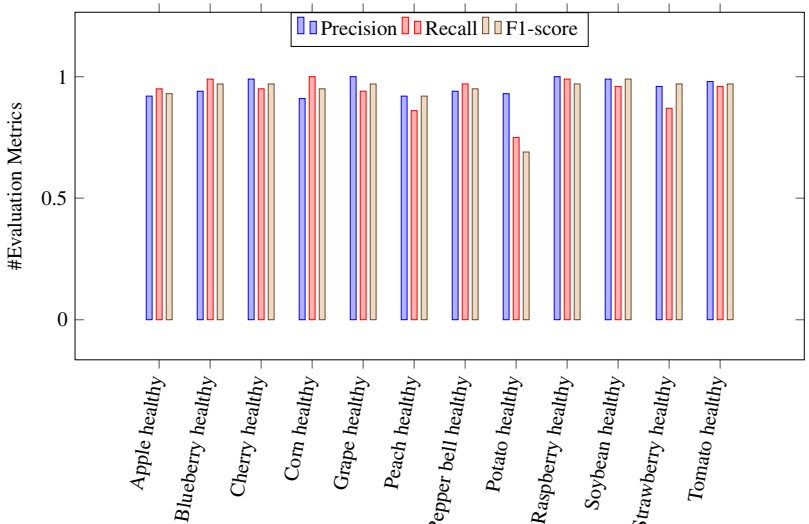

**Figure 6** Bar chart representing precision, recall and F1 score, calculated for healthy plants.

## CONCLUSION

Advancement in DL and image processing provides a prospect to extend the research and applications of detection and classification of various diseases in plants using images. In this research, simple and effective model, CapPlant is developed for classifying various categorizes of healthy and effected plants. In CapPlant, convolutional layer along with capsule layer is used to capture more features. As capsules incorporate orientation and relative spatial relationships between different entities in an object, they outperform

conventional CNN. Results obtained from experimentation clearly demonstrate the effectiveness of the proposed CapPlant model. For now, the model has been tested and validated for the publicly available PlantVillage dataset. The threats to the validity of results obtained from CapPlant model may depend upon properties such as size, unsharpness, bit depth, and noise in the underlying test images.

In the future, a recommender system with our proposed technique can be integrated to suggest various actions that need to be taken against a given disease. Moreover, the idea of using CNN with capsule networks can bring significant improvement in the performance of many already existing DL models. It can point us towards a direction to explore various applications using capsules network.

### Funding
The authors received no funding for this work.

### Competing Interests
The authors declare that they have no competing interests.

### Author Contributions
- Omar Bin Samin conceived and designed the experiments, prepared figures and/or tables, and approved the final draft.
- Maryam Omar performed the experiments, performed the computation work, authored or reviewed drafts of the paper, and approved the final draft.
- Musadaq Mansoor analyzed the data, authored or reviewed drafts of the paper, and approved the final draft.

### Data Availability
PlantVillage; An open access repository of images on plant health to enable the development of mobile disease diagnostics, *Hughes & Salath'e (2015)* obtained from *Mohanty (2018)* was used for training and testing.

-Hughes, D. P. and Salath'e, M. (2015). An open access repository of images on plant health to enable the development of mobile disease diagnostics through machine learning and crowdsourcing. ArXiv arXiv:1511.08060.

-Mohanty, S. (2018). Plantvillage-dataset. https://github.com/spMohanty/PlantVillage-Dataset

The Python file for training, all the outputs obtained during training of CapPlant, model and accuracy and loss of CapPlant, respectively, and the model files to reproduce results are available as Supplemental Files.

### Supplemental Information
Supplemental information for this article can be found online at http://dx.doi.org/10.7717/peerj-cs.752#supplemental-information.

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
