# Peer review of "CapPlant: a capsule network based framework for plant disease classification"

_PeerJ Computer Science, doi:10.7717/peerj-cs.752_

## Round 0.1 · original submission · Minor Revisions

Revise your manuscript as suggested by reviewers.

Reviewer 1 ·

Basic reporting

In the methodology section, authors have mentioned that CapPlant is a real deep learning architecture because it uses end-to-end learning. However, in research paper end-to-end learning is not explained and discussed. It would be better if the authors can explain the concept of end-to-end learning in the paper.

Experimental design

Figure 3 shows the learning curves obtained while calculating training and validation accuracy and it gives an idea that model was trained for 200 epochs. In contrast, Table 1 lists 100 epochs. Author should explain that why model was trained for 200 epochs but model till 100 epochs was used for testing and validation.

Please cite Plant Village dataset in Table 2.

Validity of the findings

It would be helpful if the authors can include the values of standard deviation values along with the accuracies reported in Table 4.

Additional comments

Overall the paper is written and organised well.

Reviewer 2 ·

Basic reporting

no comment

Experimental design

no comment

Validity of the findings

no comment

Additional comments

The authors propose a model CapPlant to classify diseases in plants using deep learning model. They try to solve this problem by incorporating capsule layer with convolutional layer. The paper is generally well written and structured. Literature review is comprehensive and authors managed to successfully discuss the importance of their research, from both a theoretical and an applied perspective. The statistical analysis been performed appropriately and rigorously. Manuscript is presented in an intelligible fashion and written in standard English. The manuscript is technically sound, and the results support the conclusions.
However, I would recommend following modifications:
1. In introduction author has mentioned that “exploiting capsule layer enables the model to capture relative spatial and orientation relationship between different entities of an object in image”. How?
2. In Table 2, author has already assigned class labels, these labels should be used in Table 4 & Table 5 instead of class names while discussing F1 score, Precision and Recall of each class.
3. Paper lacks detailed model summary of proposed CapPlant model. There should be a visual illustration that shows type of layer(input, CNN, Capsule, Dense etc ) input tensor and output tensor of a complete model.

Reviewer 3 ·

Basic reporting

The paper presents the CapPlant model to predict whether it’s healthy or contain some disease underpins deep learning based architecture. Usage of Capsule layer as the final layer convolutional neural network to include spatial information about the entities is a novel idea.

The paper provides a detailed description of the existing state of the art techniques along with their limitations.

The results for the experiments have been described in detail. The approach is showing statistically sound results.

Some Minor changes:
In Line 49-58, the authors mention that “These models have some major drawbacks, for instance one major issue with some of these models is in targeting less number of crops and diseases, secondly they are presenting results using limited or none evaluation metrices. In this research, a deep learning architecture; CapPlant is developed using CNN along with capsule network to classify and detect any disease found in plants accurately.”

It would be nice to elaborate a bit the evaluation metrics that the existing techniques are not focussing on and the ones that CapPlant is dealing with.

Experimental design

Usage of the Capsule layer as part of the CNN is quite a novel and original idea.

The process of the approach has been provided in detail. Experimental methodology and the experimental results for both the proposed approach and the existing techniques are provided. A comparative analysis is done. The approach shows statistically sound results with a better accuracy score when compared to the other approaches.

For the purpose of reproducibility, it would be nice to make available the code in the form of Git repository.

Validity of the findings

The dataset is available.

The findings of the experiments comply with the main contribution of the paper.

Threats to Validity for the CapPlant approach should be provided.

---

## Round 0.2 · Minor Revisions

1. As mentioned in the reviewer response letter, provide the github link of the source code within the manuscript.
2. Correct the English grammar of the manuscript, wherever required.

---

## Round 0.3 · accepted · Accept

The manuscript may be accepted for publication.